# The Ethiopian Magic Scrolls: A Combined Approach for the Characterization of Inks and Pigments Composition

Monia Vadrucci [1,*], Davide Bussolari [2,3], Massimo Chiari [4,5], Claudia De Rose [3], Michele Di Foggia [6], Anna Mazzinghi [4,5], Noemi Orazi [7], Carlotta L. Zanasi [3] and Cristina Cicero [8]

1    Particle Accelerator for Medical Application Laboratory, Italian National Agency for New Technologies, Energy and Sustainable Economic Development (ENEA), 00196 Rome, Italy
2    Diagnostica per l'Arte Fabbri di Davide Bussolari, 41011 Campogalliano, Italy
3    Accademia di Belle Arti di Bologna, Via delle Belle Arti 54, 40126 Bologna, Italy
4    Physics and Astronomy Department, University of Florence, Via G. Sansone 1, 50019 Firenze, Italy
5    National Institute of Nuclear Physics, Section of Florence, Via G. Sansone 1, 50019 Firenze, Italy
6    Biomedical and Neuromotor Sciences Department, Alma Mater Studiorum Università di Bologna, Via Irnerio 48, 40126, Bologna, Italy
7    Department of Industrial Engineering, University of Rome "Tor Vergata", Via del Politecnico 1, 00133 Rome, Italy
8    Department of Literary, Philosophical and Art History Studies, University of Rome "Tor Vergata", Via Columbia 1, 00133 Rome, Italy
*    Correspondence: monia.vadrucci@enea.it

**Abstract:** The Ethiopian magic scrolls are traditional parchment artifacts used by the Christians of Ethiopia as protection against disease and demonic possessions. On the occasion of their restoration in the *Accademia delle Belle Arti di Bologna* (Italy); a preliminary characterization before the treatments has been performed on four Ethiopian scrolls belonging to the *Archivio storico della provincia di Cristo Re dei Frati Minori dell'Emilia Romagna* of Bologna (Italy). In order to plan an effective preservative restoration procedure and; at the same time; to investigate the manufacturing techniques; the text and the decorations on the magic scrolls were studied and characterized. A combined approach by imaging and compositional techniques was used: Infrared Reflectography (IRR) for the preliminary characterization of the graphic supports and the identification of the points to sample the chemical measurements; and the spectroscopic analyses to clarify the hypothesized investigations and confirm the chemical composition of the inks. In particular; Attenuated Total Reflectance-Fourier Transform Infrared (ATR-FTIR) spectroscopy has provided information relating to the molecular composition of inks and pigments; while a characterization of the constituent elements is obtained with the Ion Beam Analysis (IBA). The ink composition proved to be consistent with data generally documented in the literature and contributing to the expansion of knowledge on Ethiopian magic scrolls and their production.

**Keywords:** magic scroll; mixed ink; infrared reflectography; ATR-FTIR spectroscopy; ion beam analysis

## 1. Introduction

The Ethiopian magic scrolls, also called *yabrānnā kətāb*, are parchment artifacts used by the Christians of Ethiopia as protection against disease and demonic possessions. They generally consist of one or more bands of parchment joined together by a thin strip of leather. They are still produced and used by the Ethiopian population but, since the 1970s, their use has undergone a progressive decline [1].

The production of this kind of artifact has a centennial tradition probably born during the reign of Aksum (I-VII b. C.). Despite the wide use of this type of artifacts over time, no specimens dating back to the 18th century still exist today, probably because they are produced for daily use, leading to their rapid deterioration and consequent loss. The Christian tradition tends to attribute the scarcity of ancient manuscripts before the

16th–17th century to the destruction of the Christian cultural heritage pursued in the first half of the 16th century by the Emir Ahmad ibn Ibrahim al-Ghazi [2]. Indeed, wars and devastation must have seriously affected the Ethiopian manuscript heritage, although the extension of this destruction cannot be precisely determined.

These objects, similar to amulets, arise from the synthesis between ancient local pagan beliefs and Christianity, the dominant monotheistic religion since the 4th century AD. They report "magical-religious" texts with a protective or curative function for the owner. They are not meant to be recited or read, but their effectiveness lies only in their writings, thanks to which the scrolls acquire healing powers.

The text contained in these scrolls is usually drawn in red and black ink; it is generally composed in the gəʿəz language, often mixed with Amharic or Tigrinya, also borrowing some terms from Greek, Hebrew, and Arabic.

The textual parts are interspersed with drawings of protective angels (usually located at the beginning of the scroll) or other talismanic figures, including demons, imprisoned devils, and human faces enclosed in stars or combinations of squares, with the recurring "all-seeing eye" symbol. The spell written in a magic scroll is considered more effective than a verbal spell precisely because the magical power is amplified by the presence of these graphic elements and talismanic drawings [3].

The Ethiopian scrolls were produced for personal use, and that is why the owner's name is repeated within the text several times; during the manufacturing, even the parchment strips that made them up were cut to a length corresponding to the height of the owner to protect him "from head to toe". In the case of a change of ownership, for example, the old owner's name was erased and replaced with the new one [4].

As a preliminary characterization before the restoration of a cultural heritage artifact, the analysis of inks and pigments employed to write the handwritten text is of particular interest. Identifying their composition is a fundamental step to reconstructing the history of the investigated artifact and, as in the case of such unique artifacts, how they fit into a traditional production process rooted and standardized over time. Furthermore, the precise knowledge of the materials employed in the manufacturing process of the investigated artifact is also of fundamental help in defining the ideal conservation approach for the object and, in case it requires restoration treatments, it allows for the design of a targeted intervention effective in restoring the functionality of the object and respectful of its composition and peculiarities.

In such a context, imaging techniques have proven to be a valid aid for the preliminary characterization of the graphic media employed in the graphical and handwritten artifact, giving general and introductory information on the investigated text and acting as a guideline for subsequent punctual analyses. Through such techniques, it is possible to group pigments and inks with similar spectral behaviors obtaining a rough indication of their possible composition. On the other hand, the analysis of the elemental composition of the writing media allows their more precise characterization confirming the indications obtained with the preliminary imaging investigations or resolving any ambiguities due to, for example, the use of mixtures of different inks or pigments.

This combined approach is more valuable in the case of handcrafted artifacts with uncertain dates and placed in a traditional production framework that spans centuries, as in the case of Ethiopian magic scrolls. The maintenance of the iconographic and paleographic tradition unchanged for centuries, as well as the production technique of the parchment support, makes their historical collation difficult since the ancient specimen often is similar in appearance to the ones produced in the contemporary time.

In this work, we present the results of the different analysis performed on four Ethiopian magic scrolls belonging to the *Archivio storico della provincia di Cristo Re dei Frati Minori dell'Emilia Romagna* of Bologna (Italy) on their restoration in the *Accademia Delle Belle Arti* of Bologna (Italy).

A preliminary characterization has been performed using near Infrared Reflectography (IRR), thanks to which it has been possible to obtain an indication on the pigments and

inks composition based on their response in the reflectograms. This first phase addressed the choice of the subsequent techniques to obtain the most concrete result and choose the points to carry out the measurements.

The spectroscopic analyses clarified the indications obtained with the preliminary imaging investigation and were useful in resolving ambiguities due to the use of mixed inks. In particular, through the Attenuated Total Reflectance-Fourier Transform Infrared (ATR-FTIR) spectroscopy, it has been possible to obtain information related to the molecular composition of the ink/pigments. Finally, the Ion Beam Analysis (IBA) has helped in the elemental characterization of the investigated writing media.

## 2. Materials and Methods

### 2.1. The Ethiopian Magic Scrolls

The four Ethiopian scrolls under study belong to a small collection of ethnic materials from Ethiopia and they are preserved in the *Archivio storico della provincia di Cristo Re dei Frati Minori dell'Emilia Romagna* of Bologna (Italy). Three of the four finds, before being transferred to the Archive, were kept in the *Museo delle Grazie* in Rimini where they had been marked with the inventory numbers 642 (Figure 1a), 643 (Figure 1b), and 644 (Figure 1c). The exact period in which the scrolls were part of the *Museo delle Grazie* is not known. However, they were inventoried in 1969. The fourth scroll (Figure 1d), on the other hand, does not seem to come from the *Museo delle Grazie* as it has no inventory number and has never been mentioned in the lists of museum assets. In relation to it, a photocopy of an annotation preserved together with the manuscripts was found; this would document the donation of roll at the Bologna Archive in 2005. The photocopy bears the following inscription:

"Trovato in un/Tukul d'un ras
da/Vespignani Francesco/in Adis
Abeba nel 1937/Vespignani
Giovanni/Via S. Corbari 37/Forlì/Tel
68287/in archivio a BO/27/8/05".

### 2.2. Imaging Techniques

The multispectral analysis is commonly employed for the study of techniques and materials used and to determine the preservation state of paper or parchment artifacts. The procedure is based on the evaluation of the response that the various materials offer when irradiated by different light sources operating on wavelengths longer or shorter than the visible band; on the other hand, the procedure is also based on the tonal rendering that the same materials assume when observed with instruments sensitive to a wider spectrum than the perception capabilities of the human eye.

IR Reflectography

The infrared images were acquired by means of a Sony Cyber-shot DSC-F828 camera with 2/3″ (3.9×) CCD sensor and a resolution of 8.0 megapixels. The camera is combined with a Carl Zeiss 7.1–51.0 mm f/2.0–2.8. Different filters (Hoya R72 Infrared Filter, B+W 093 RG830 Filter, Schott RG1000 Filter) were applied in succession to the lens to determine the variation of the behavior of the inks with the variation of the passing infrared band: the evaluation is based on the different opacity or transparency infrared degrees depending on the ink formulation, recognized using different filters. For this purpose we proceed progressively, starting the acquisition without any optical filter and then increasing the length filtering 100 nm per step, from 700 nm up to 1000 nm. The images were acquired in TIFF format and processed using the Adobe Camera Raw and Adobe Photoshop software, optimizing the contrasts to gain in both light and dark parts, maintaining the same settings for all shots in order to obtain comparable results.

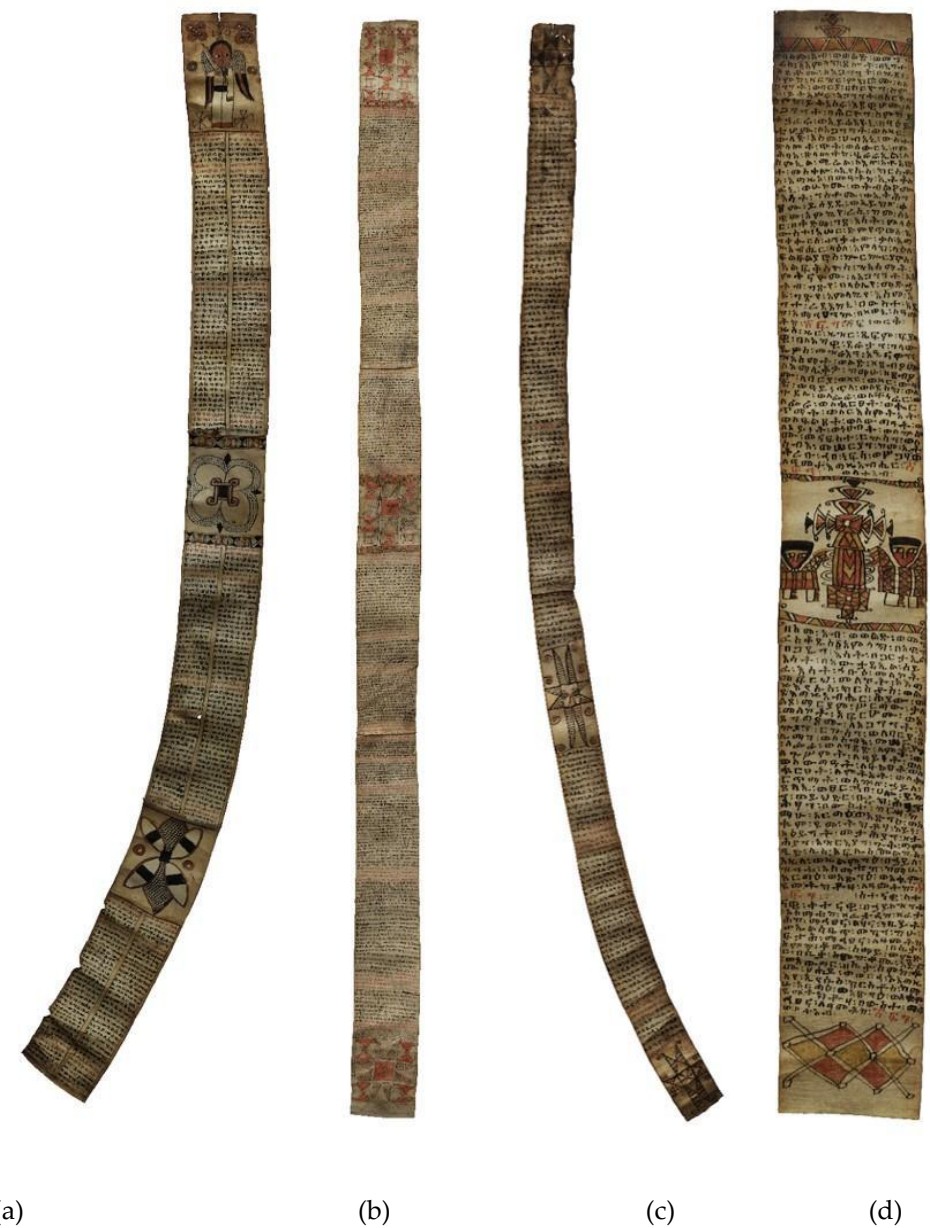

(a) (b) (c) (d)

**Figure 1.** Ethiopian magic scrolls: (**a**) num. 642 (1706 mm × 127 mm); (**b**) num. 643 (2097 mm × 127 mm); (**c**) num. 644 (1600 mm × 67 mm); (**d**) without number (644 mm × 85 mm).

*2.3. ATR-FTIR Spectroscopy*

Spectral analyses were performed using a Bruker ALPHA Platinum ATR spectrometer provided with a single reflection diamond in Attenuated Total Reflectance (ATR) module; the spectra were collected in the 4000–500 cm$^{-1}$ range with a resolution of 4 cm$^{-1}$; each spectrum was the average of 64 measures. The surveys were carried out on the points indicated in Figure 2. It is worth noticing that to maximize the information obtainable from the ATR-FTIR spectroscopy, it was necessary to isolate, within the spectra, the bands attributable exclusively to the inks/pigments. To achieve this, it was essential to subtract the contribution of the underlying parchment support from the inks' spectra: by eliminating the bands relative to the parchment from the spectrum, it was possible to identify the inks' components.

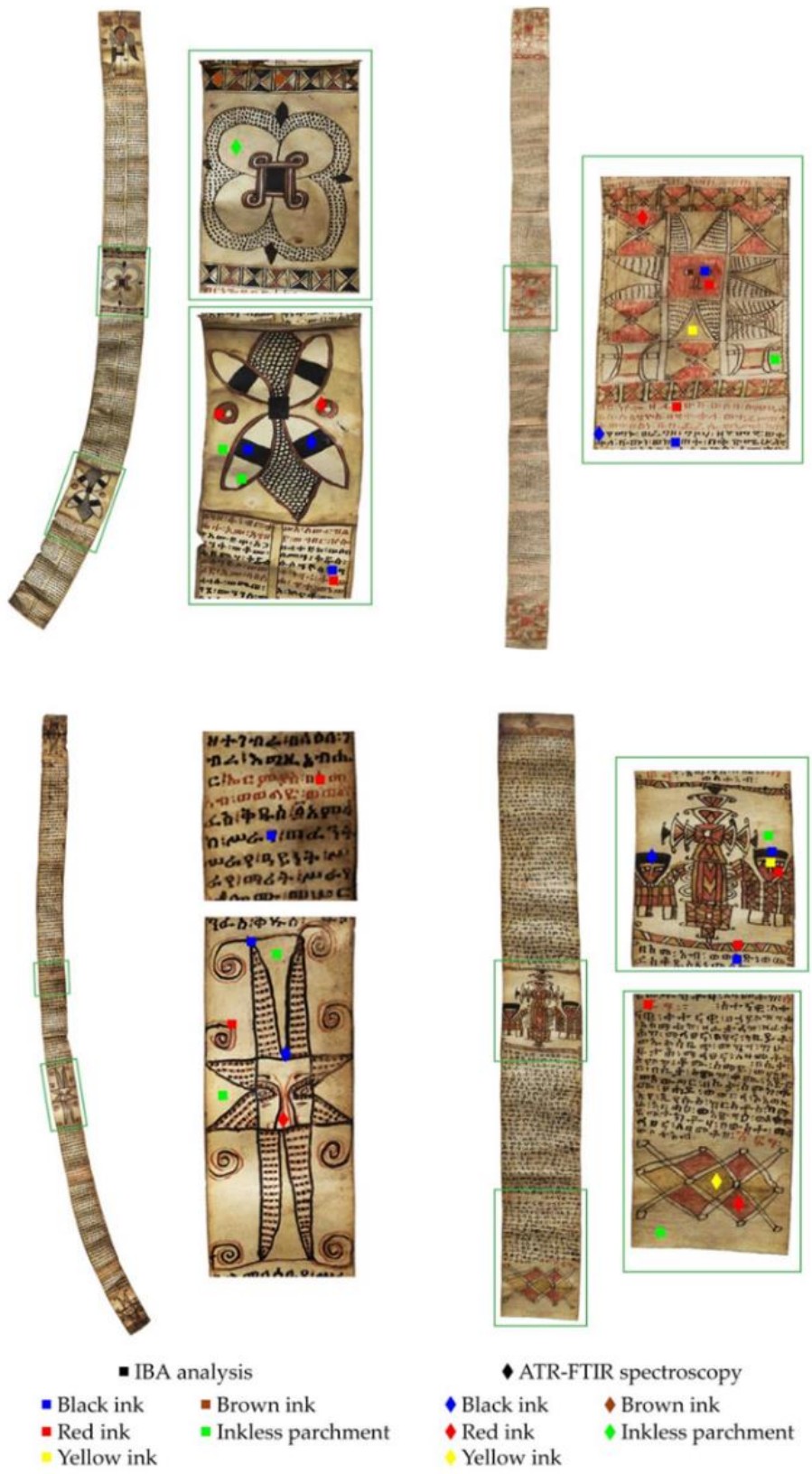

**Figure 2.** Photograph of Ethiopian magic scrolls where the points investigated by ATR-FTIR spectroscopy and IBA analysis were marked.

*2.4. IBA*

Ion beam analysis measurements were performed at the external beamline for cultural heritage applications of the 3 MV Tandetron accelerator of INFN LABEC laboratory in Florence [5]. Under the name of IBA techniques belong several analytical methods, of which the following are employed at the cultural heritage beam line at LABEC: Particle-Induced X-ray Emission (PIXE), Particle-Induced Gamma-ray Emission (PIGE), and Elastic Backscattering Spectroscopy (EBS). These techniques are performed simultaneously, following the so-called "Total-IBA" approach [6]. These analytical methods, employed together, allow a comprehensive characterization of the elemental composition and depth distribution of the analyzed material. The set-up at the end of the beamline includes two X-ray detectors for PIXE technique: one is a 10 mm$^2$ Silicon Drift Detector (SDD) with He flow for light and major elements analysis, and the other is a 150 mm$^2$ SDD, with a 450 μm thick Mylar absorber for heavy and trace elements; one gamma-ray detector for PIGE technique, a 20% relative efficiency HPGe detector with a mechanical cooler; and one particle detector for EBS technique, a Si pin diode 10 × 10 mm$^2$ active area, placed at 135° scattering angle and mounted in an aluminum case, kept at $10^{-1}$ mbar pressure. Measurements were carried out on the points marked in Figure 2 using a 3 MeV proton beam, 0.5 mm in diameter, extracted into ambient pressure through a 200 nm thick $Si_3N_4$ window, using a beam current ranging from 100 to 500 pA (chosen to keep dead time and pile up corrections negligible), and lasted 300 seconds. The measurement set up is presented in Figure 3.

Quantitative results were then obtained by accurate charge-equivalent normalization measuring the weak extracted beam currents using a rotating chopper [7]. Quantitative IBA results based on the analysis of PIXE spectra were carried out using the GupixWin software [8], whereas the analysis of the EBS spectra was achieved using the SIMNRA software [9]. The analysis of PIGE spectra, typically used to determine light elements such as Li, Be, B or F, did not reveal the presence of these elements.

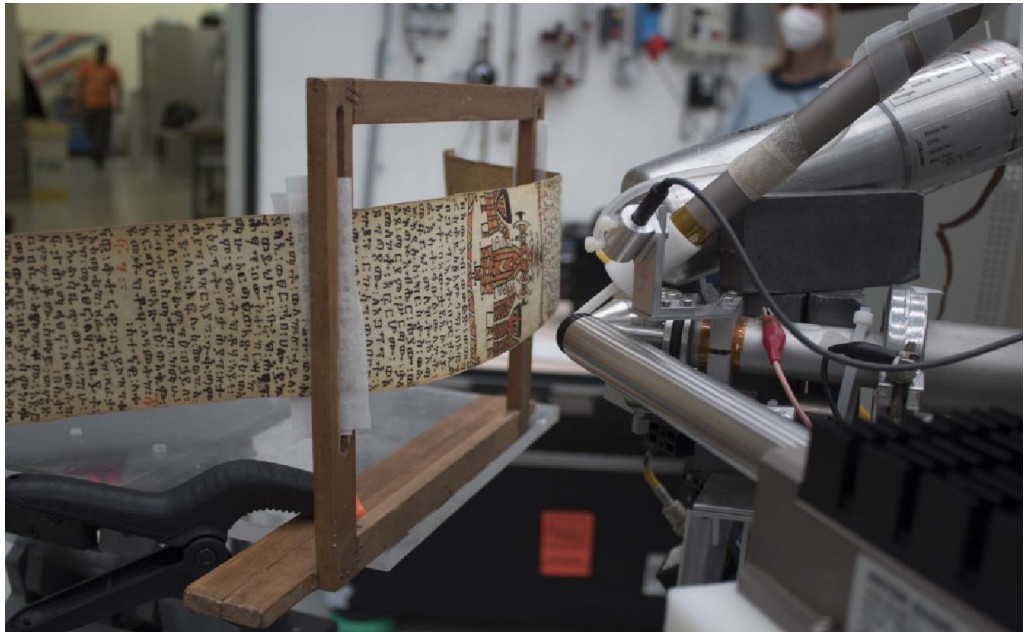

**Figure 3.** Ethiopian scroll without number during the IBA measurement at the external beam line of INFN LABEC.

## 3. Results

*3.1. Imaging Investigations: IRR*

For a preliminary characterization of pigments and inks of all the scrolls, they have been investigated through IRR. The techniques allow for having some indication of the ink and pigment composition thanks to the possibility of evaluating their behavior in the

near-infrared (NIR) spectral range. Some pigments can absorb light and be opaque in the IRR image obtained, while others become transparent, revealing the underlying support. Generally, black carbon-based pigments derived from combustion, such as bone black or smoky black, tend to absorb light and intensify their black color in the NIR spectral range [10]. In contrast, other pigments, such as red and yellow or metal-gall inks, are transparent to IR [11,12].

Thanks to the capability of some kinds of pigments to be transparent to IR radiation, this technique is often used to highlight the possible presence of a preparatory drawing [13,14]. In this specific case, it has been employed to obtain preliminary information on possible differences in the inks and pigments composition.

Generally, black carbon-based pigments derived from combustion, such as bone black or smoky black, tend to absorb light and intensify their black color in the NIR spectral range. Conversely, iron-gall or plant-derived compounds are more transparent at these wavelengths, so they tend to disappear in the image. This phenomenon is less evident the higher the percentage of ferrous component or insoluble pigment in the mixture.

Red and yellow inks, when subjected to infrared radiation, normally show a significant drop in opacity, as they are frequently of vegetable origin.

As far as black inks are concerned, the evaluation of the images leads to a rough definition of the carbonaceous material content. Black inks hardly appear in the form of pure substance: generally, they are mixtures produced by the combination of liquids carbonaceous or iron-gallic origin, present in different quantities on the basis of the cultural background or the resource availability at the time of manufacture.

Therefore, the infrared images allow a comparison of the inked surfaces on the basis of the degree of gray tone induced by the percentage of carbon.

The formulations characterized by a predominance of substances of iron-gallic or vegetable origin tend to assume a certain transparency in infrared reflectography, more or less marked on the basis of the variety and concentration of the components making up the mixture.

At first, from the reflectographic images (Figures 4–6), it can be possible to notice that the response of the red and yellow pigments of all four scrolls is the same behavior. They are completely transparent to the IR radiation showing the underlying parchment.

Differently for the black inks and pigments, there are differences among the scrolls. It is possible to notice that for the scroll without number and for scrolls no. 642 and no. 644, we had a similar response with the persistence of the written text also in the reflectographic image. At this range of wavelengths, it is impossible to affirm the ink's nature since the results obtained in the NIR range only suggest the possibility of the presence of a carbon-based compound [15,16].

On the contrary, the image of scroll no. 643 (Figure 7) shows the different behavior of both the ink of the writing and of the drawings that partially disappear, leaving the underlying parchment visible and suggesting the possibility of the use of a mixed ink composition based on a metal-gall ink with some carbonaceous contamination. This hypothesis is further suggested by the naked eye observation of its color, which results in a tendency to be more brown than the others, reflecting a typical feature of metal-based inks [17].

The IRR technique did not allow for a certain attribution of the type of inks and their composition, but it did allow us to definitely establish that scroll no. 643 has a black ink with different spectral features than the other specimens. This behavior can be attributed to a different chemical composition as well as, for example, to a different dilution of the compound without giving any assurance on their nature.

Interestingly enough, thanks to the IR images, it was also possible to notice that the red and black inks and pigments employed to write the text and trace the drawings on the scroll had the same infrared response, thus confirming the documented Ethiopian practice of using the same ink both for the written part and for the decorative apparatus [18].

The analysis therefore did not identify any discrepancies attributable to out-of-context tampering, additions, or restoration work.

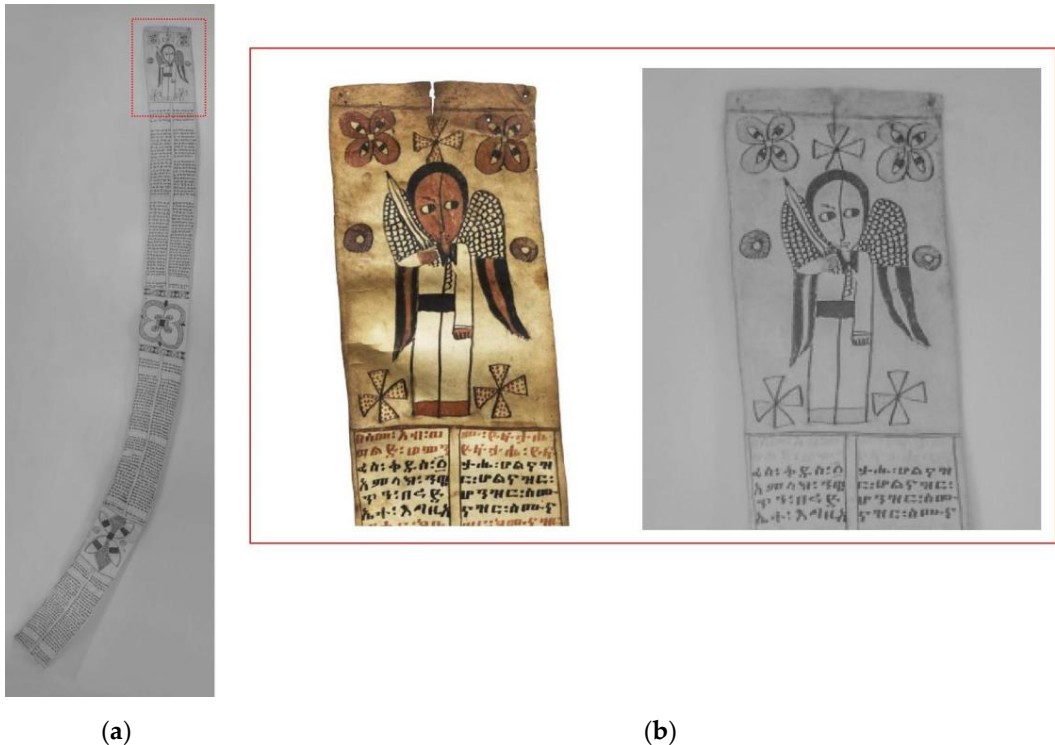

**Figure 4.** Ethiopian scroll no. 642: (**a**) IR reflectographic image of the entire scroll; (**b**) magnification of the IR reflectographic image and the corresponding visible one of part of the decoration and the writing.

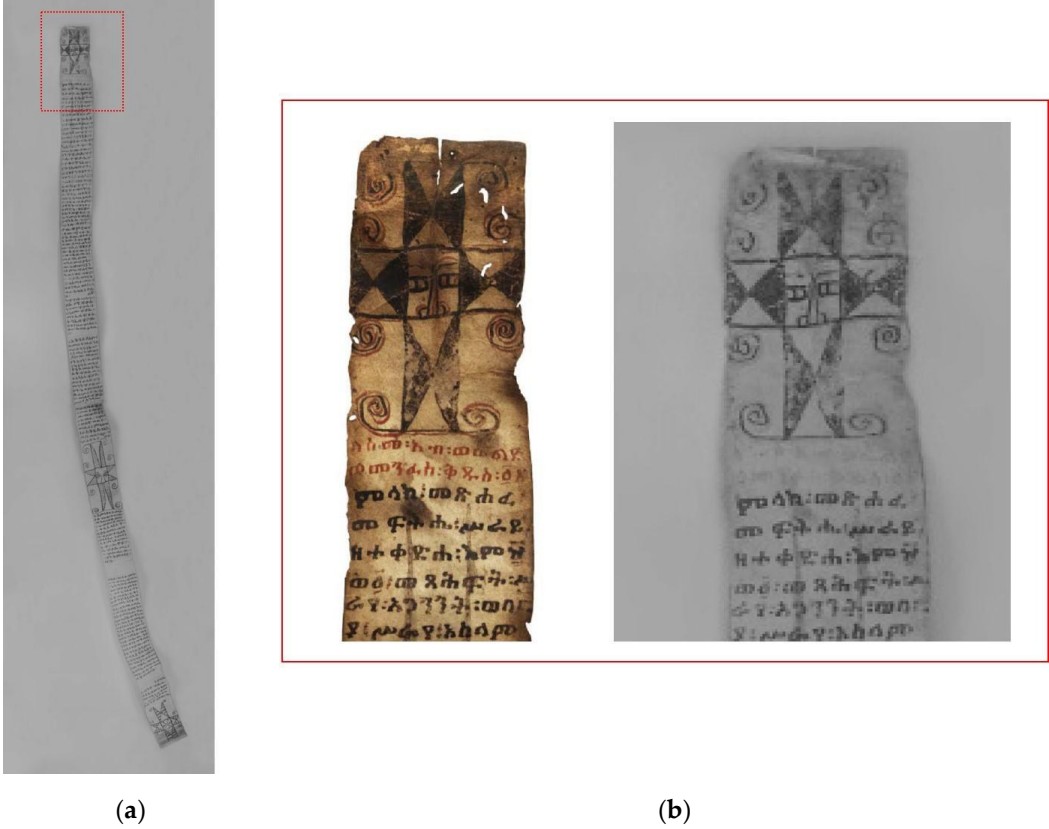

**Figure 5.** Ethiopian scroll no. 644: (**a**) IR reflectographic image of the entire scroll; (**b**) magnification of the IR reflectographic image and the corresponding visible one of part of the decoration and the writing.

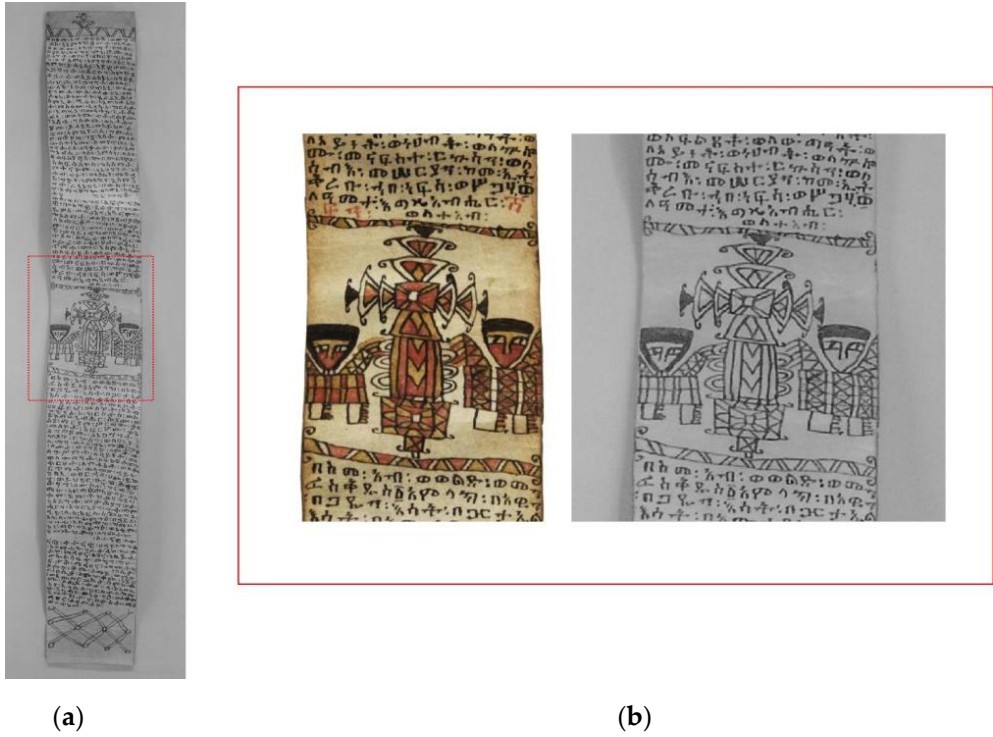

(**a**)                                                                                       (**b**)

**Figure 6.** Ethiopian scroll without number: (**a**) IR reflectographic image of the entire scroll; (**b**) magnification of the IR reflectographic image and the corresponding visible one of part of the decoration and the writing.

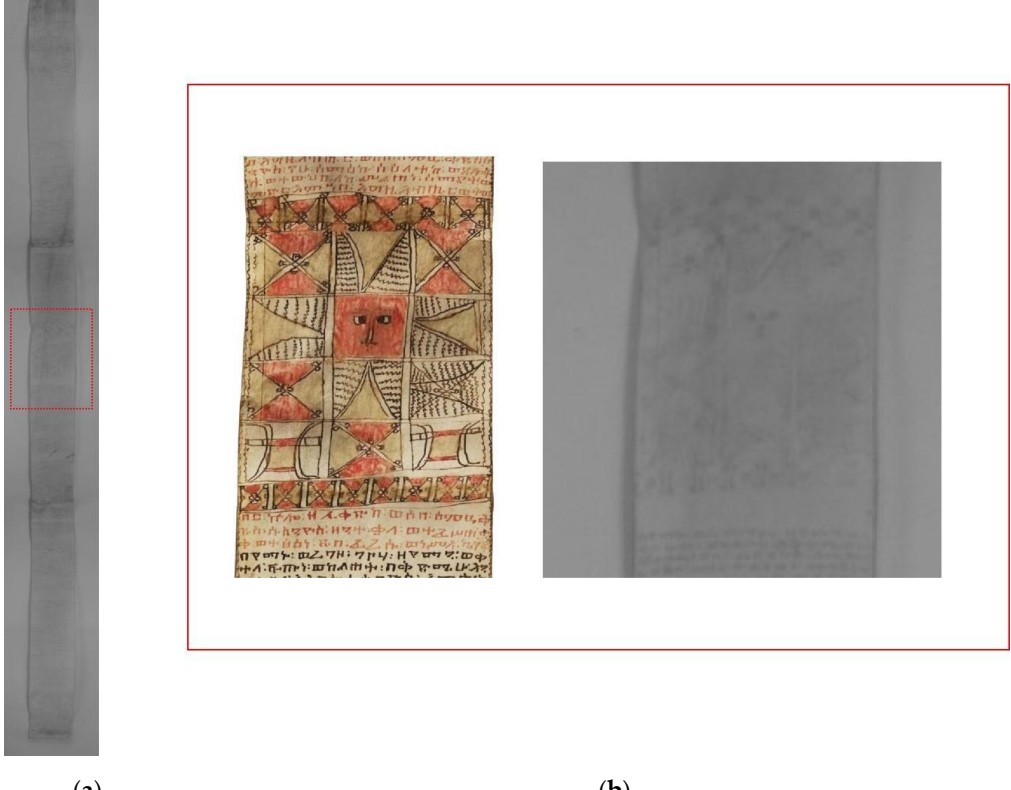

(**a**)                                                                                       (**b**)

**Figure 7.** Ethiopian scroll no. 643: (**a**) IR reflectographic image of the entire scroll; (**b**) magnification of the IR reflectographic image and the corresponding visible one of part of the decoration and the writing.

### 3.2. ATR-FTIR Spectroscopy

### 3.2.1. Inkless Parchment

The ATR-FTIR analysis performed on an inkless area of scroll no. 642 (Figure 8) showed the presence of peaks attributable to carbonates (1420 cm$^{-1}$), silicates (1015, 913, 776, 664, 527, and 463 cm$^{-1}$), and aluminates (823, 440 cm$^{-1}$) [19]. The carbonates could be linked to the residues of the use of calcitic substances, such as hosā [2], employed, according to the Ethiopian tradition, for processing the flesh side of the parchment. Alumino-silicates are instead probably due to kaolinite ($Al_2Si_2O_5(OH)_4$), the silicate mineral at the base of kaolin. These compounds can be due to a specific step of the Ethiopian parchments' preparation. In traditional manufacturing procedures, in fact, to prepare the surface for writing, the parchment was treated with clay shards called *madmat* or *madmas*, aiming to facilitate the ink's absorption [2].

IR spectroscopy revealed the presence of gypsum (1611, 1098, 664, 602 cm$^{-1}$) [19] and organic material, i.e., a resin (bands at 1710, 1359, 1252, and 1161 cm$^{-1}$) which showed a certain degree of degradation highlighted by the presence of oxalate peaks (1312 and 575 cm$^{-1}$) [20].

### 3.2.2. Black Ink/Pigment

Numerous bands attributable to phenols have been detected in the scroll's spectra. Figure 8 shows the comparison among the black inks of scrolls no. 642, no. 643, no. 644 and without number, together with the attribution of the main bands. The bands attributed to phenols are 1606–1593 (mixed with Arabic gum bands), 1484, 1363–1359, 913–907, 776 (combined with silicate vibrations), and 668 cm$^{-1}$ [20,21]. Phenols, and more aromatic substances in general, can be found indiscriminately both in carbonaceous inks and in metal-gallic inks; therefore, the presence of this chemical compound does not allow us to identify univocally whether it is one kind of ink rather than the other or, possibly, a mix of the two [22–24].

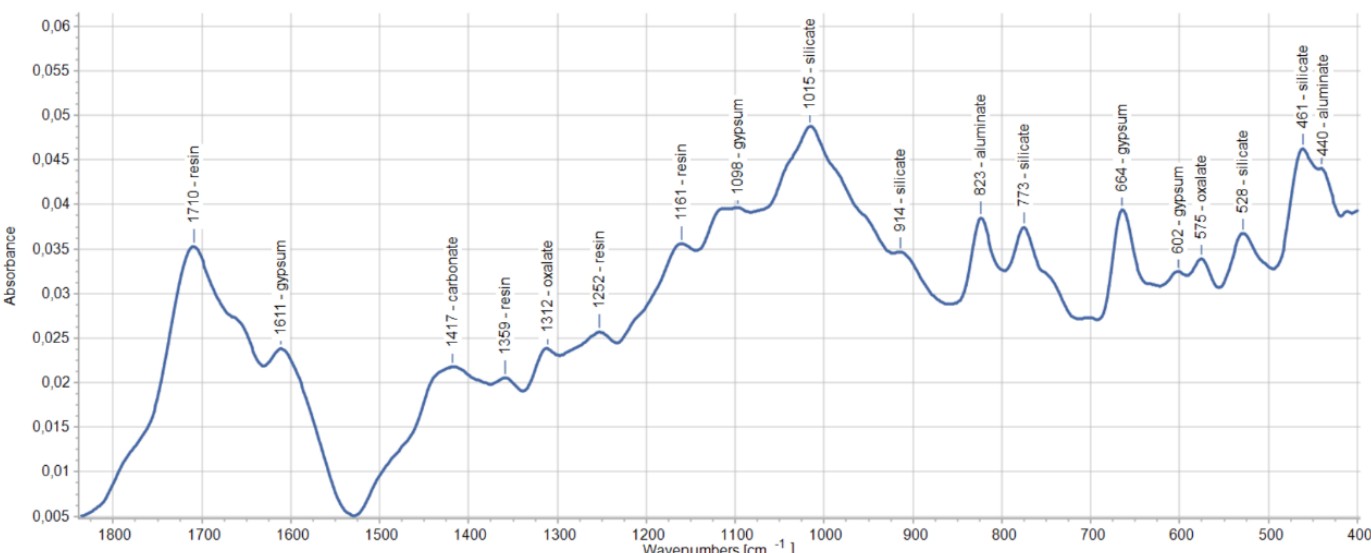

**Figure 8.** ATR-FTIR spectra of an inkless part of the black inks on scroll no. 642 with the attribution of the main bands.

The bands at 1735–1727 cm$^{-1}$ in scroll no. 642 and without number are indicative of the effects of metal-gallic inks on parchment, which causes the oxidative hydrolysis of the proteinaceous support [25]. The binding medium, identified as a gum, showed typical bands at 1603–1593, 1417 (mixed with carbonates), 1073, and 1008–987 cm$^{-1}$ [21]. Silicate bands at 913–907, 776, 688–686, 519, and 459–456 cm$^{-1}$ and carbonate bands at 1419–1413 and 876 cm$^{-1}$ as previously described for the inkless parchment [19]. Carbonates' bands could also indicate carbonaceous inks [22]. The presence of oxalate bands at 1319–1316 cm$^{-1}$ indicates gum degradation processes. The large sulphate band at 1108 cm$^{-1}$ is caused by using metal sulphates (iron, copper) and potash (potassium and aluminum sulphate) to prepare metal-gallic inks [21,22]. As it is possible to see in Figure 9, in all the different spectra the bands remain essentially the same, but those of scroll no. 643 show decidedly different intensities. The latter could indicate the presence of ink with a different formulation, although based on very similar chemical components. In particular, bands attributed to the gum (1603, 1073 cm$^{-1}$) and its degradative products (i.e., oxalates) showed the highest intensity, suggesting a higher binder content.

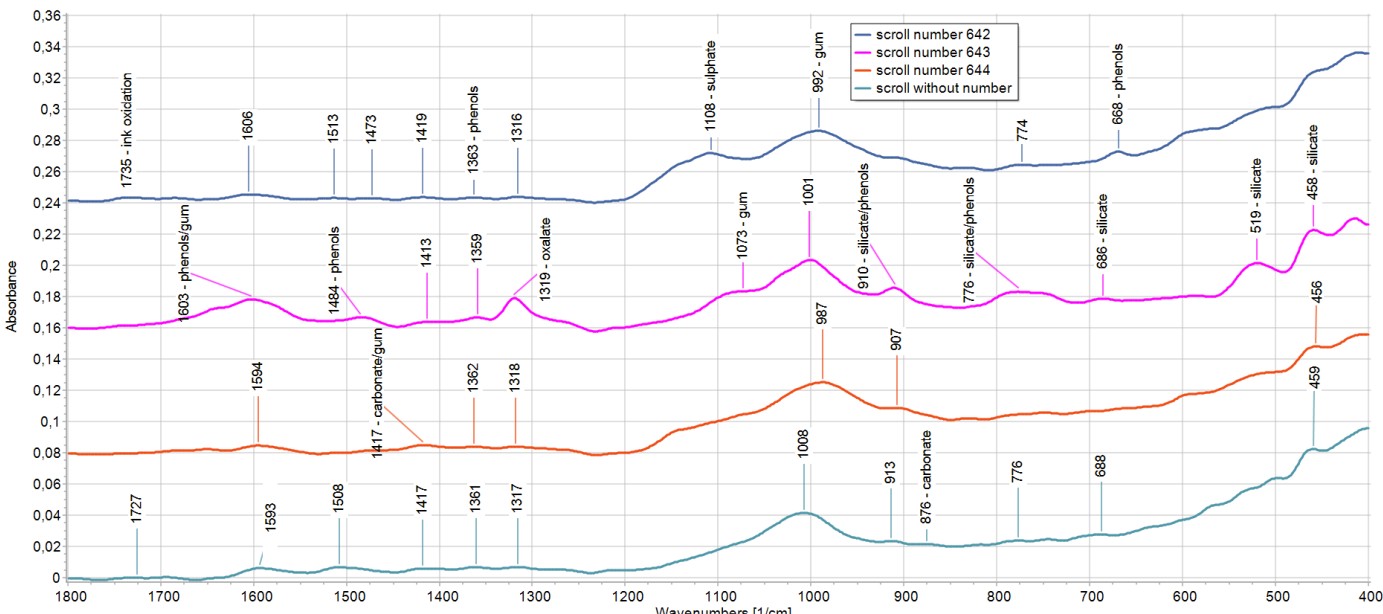

**Figure 9.** ATR-FTIR spectra of the black inks on scrolls no. 642, no. 643, no. 644 and without number, with the attribution of the main bands: to increase the readability of spectra, the attribution is reported only once, usually in correspondence to the most intense peak (i.e., the 1603 cm$^{-1}$ band of phenols is written only for scroll no. 643).

### 3.2.3. Red Ink/Pigment

Numerous bands attributable to silicates (910–908, 778, 687, 527–522, and 462–459 cm$^{-1}$) have been recorded in all the red inks/pigments of the investigated Ethiopian scrolls (Figure 10). Silicates are a class of minerals known to be, together with iron oxides (large band at 514 cm$^{-1}$), among the main components of earths and ochres [10]. Accordingly, an earthy origin can be determined for the pigments used for the red inks combined with a rubber-based binder (bands at 1624, 1070, 1001–996 cm$^{-1}$) [21].

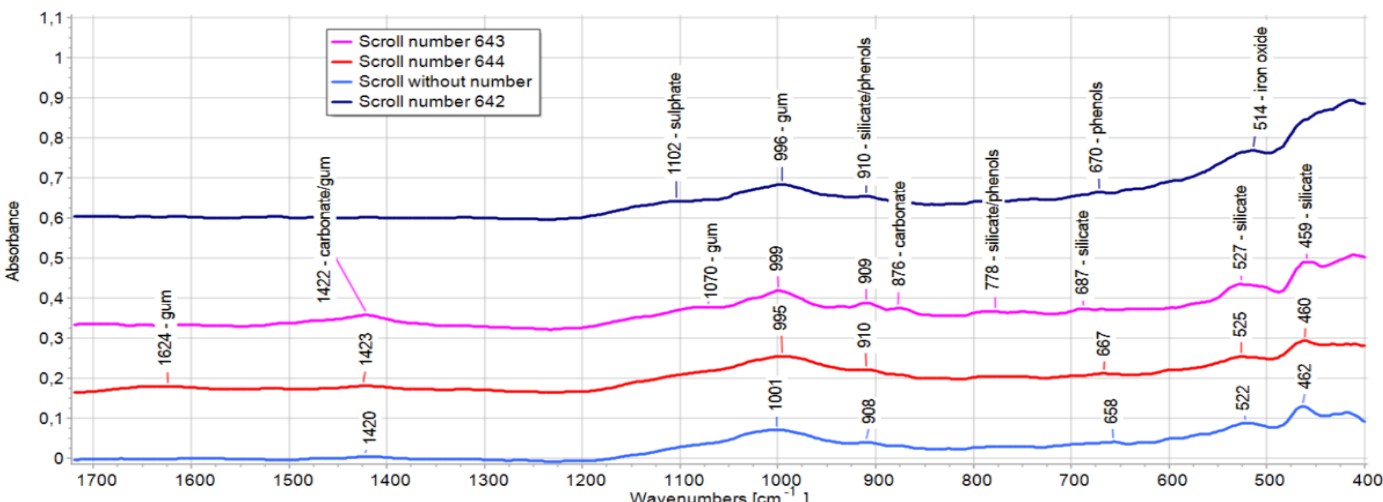

**Figure 10.** ATR-FTIR spectra of the red inks on scrolls no. 642, no. 643, no. 644 and without number, with the attribution of the main bands: to increase the readability of spectra, the attribution is reported only once, usually in correspondence to the most intense peak (i.e., the 996 cm$^{-1}$ band of gum is written only for scroll no. 642).

### 3.2.4. Yellow Ink/Pigment

The yellow pigment of the scroll without number presents several peaks (1260, 827, and 740 cm$^{-1}$) attributable to a resinous material of vegetal origin, i.e., saponins (Figure 11) [26]. The presence of intense bands attributable to silicates (912, 769, 520, and 464 cm$^{-1}$), as previously described for the red ink, could highlight the presence of an earth/ocre pigment [27].

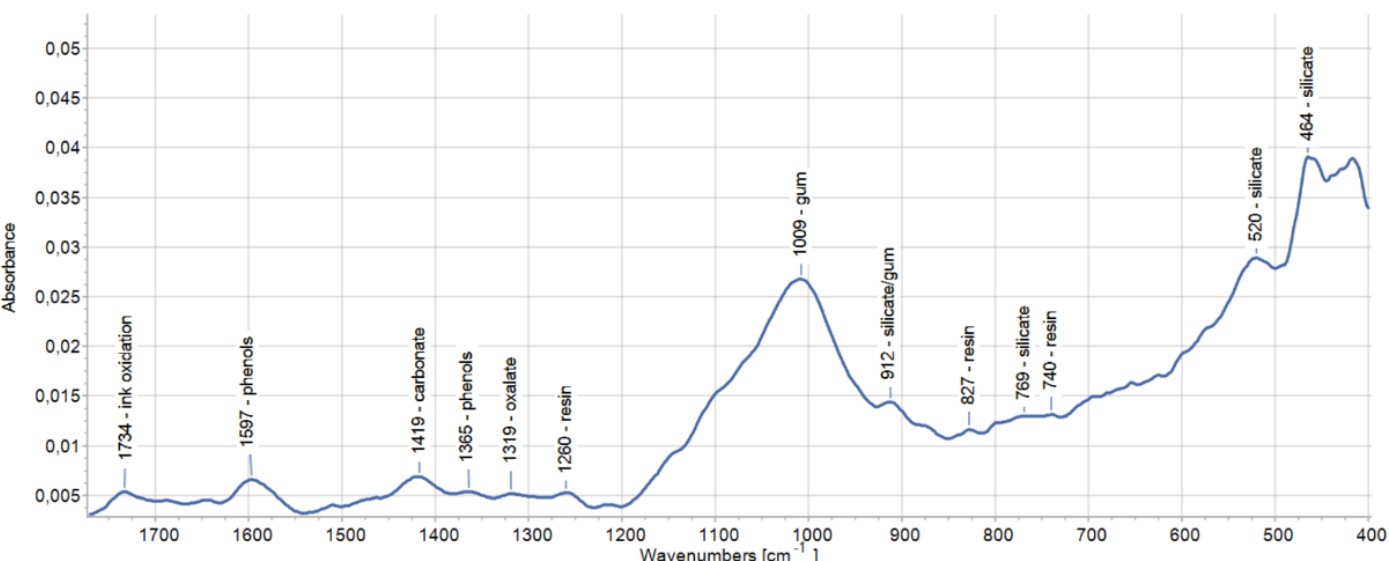

**Figure 11.** ATR-FTIR spectra of the yellow pigment on the scroll without number, with the attribution of the main bands.

Also, in this case, a rubber-based binder is employed, as highlighted by the large band at 1009 cm$^{-1}$ [21].

### 3.2.5. Brown Ink/Pigment

The brown ink of scroll no. 642 (Figure 12), such as the black inks, showed the presence of bands attributable to phenols (1596, 1512, 1369, and 912 cm$^{-1}$) [21]. Therefore, it is not easy to establish whether it could be a carbonaceous ink or a metal-gall one. In this sense, the higher intensity of carbonates' bands at 1416 and 872 cm$^{-1}$ compared to previous samples may suggest that this ink formulation is mainly of carbonaceous origin [22]. The presence of silicate bands (912, 767, 684, 520, and 461 cm$^{-1}$) could also indicate the presence of an earth-based pigment [27].

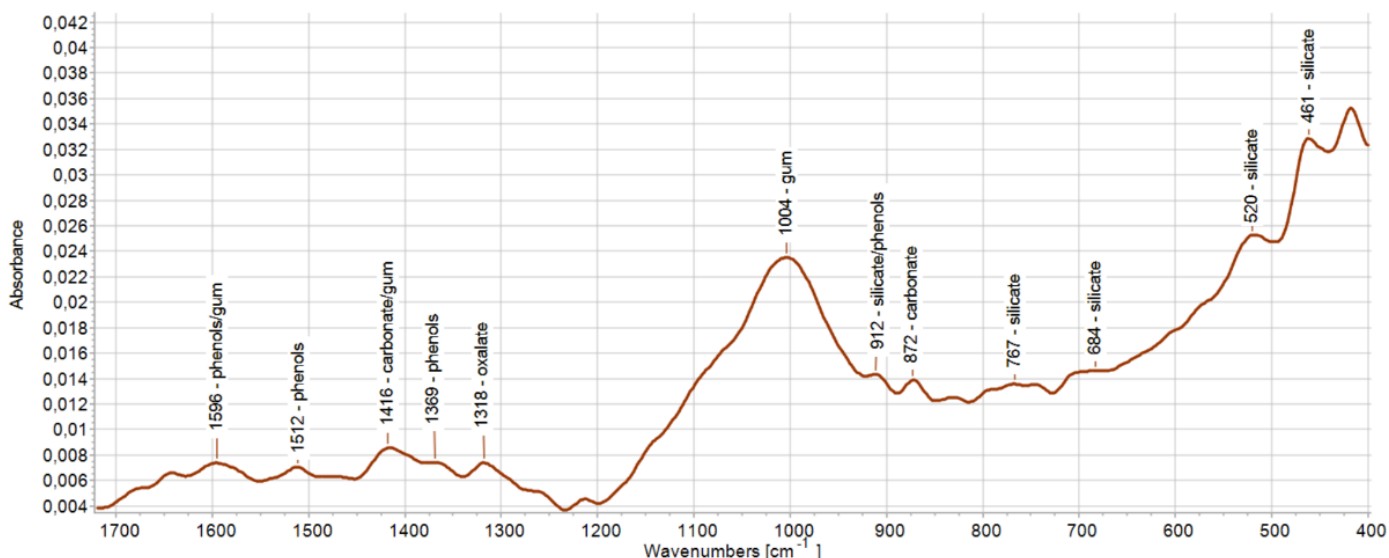

**Figure 12.** ATR-FTIR spectra of the brown pigment on scroll number 642, with the attribution of the main bands.

### 3.3. IBA Analysis

Table 1 summarizes all the main elements found within the inks and pigments of the investigated Ethiopian scrolls. In the following paragraphs, the composition of the writing media and the relative attributions are discussed in detail.

**Table 1.** Elemental analysis of inks and pigments obtained by IBA.

| Ethiopian Scroll | Ink/Pigment | Elements |
|---|---|---|
| Scroll without number | Parchment | Ca, K, Fe, Cl, Si, (Cu), (Zn), (Br), (Hg) |
| | Black | Fe, Hg |
| | Red | Hg, Fe, S, (Ti) |
| | Yellow | Fe, Hg |
| Scroll no. 642 | Parchment | Ca, K, Fe, Br, Cl, (Cu), (Zn) |
| | Black | Ca, K, S, Fe, (Cu) |
| | Red | Ca, Fe |
| | Brown | Ca, Fe, K |
| Scroll no. 643 | Parchment | Ca, K, Fe, Cl, (Hg), Cr |
| | Black | Ca, K, Fe |
| | Red | Hg, S, Fe |
| | Yellow | Fe |
| Scroll no. 644 | Parchment | Ca, K, Fe, Cl |
| | Black | Ca, K, Fe, (Ti), (Zn), (Pb) |
| | Red | Hg, S, Ca, K, Fe |

### 3.3.1. Black Ink/Pigment

Black inks are characterized by the presence of calcium (Ca), sulfur (S), and potassium (K), elements that may suggest the use of an organic compound. Nevertheless, in all four scrolls iron (Fe) was also detected in variable amounts and may indicate the use of iron-gall ink (Figure 13). Therefore, we are dealing with mixed carbonaceous/metal-gallic inks in all four scrolls. These custom results to be attested in Ethiopia have been reported in the XRF analysis of the codices belonging to the collection of the church of ʿUra Mäsqäl [28].

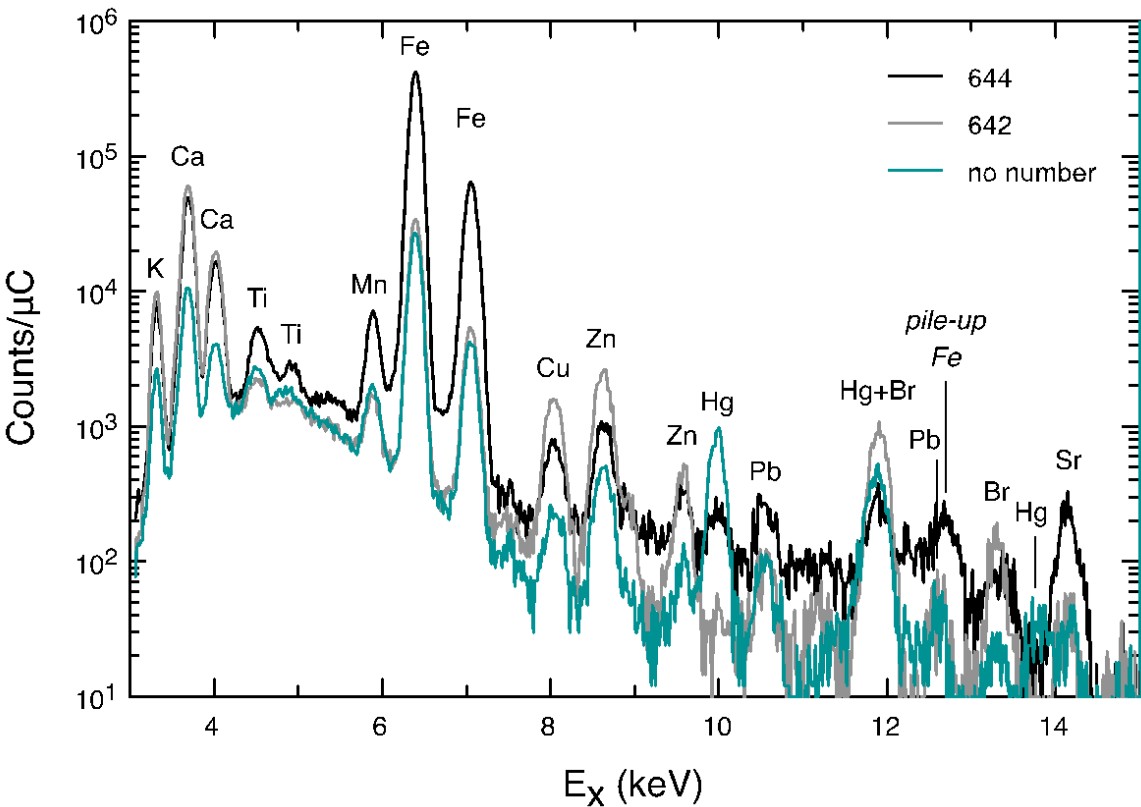

**Figure 13.** Comparison of PIXE spectra of the black inks on scrolls with no number, no. 642, and no. 644 obtained with the large area SDD.

Moreover, the presence of potassium could be traced back to the use of a gummy binder, more specifically, Arabic gum, in which the ink was usually dispersed to be used for writing [2].

### 3.3.2. Red Ink/Pigment

The analysis on red inks of scrolls no. 643, no. 644, and of the scroll without number (Figure 14) reveals a large presence of mercury (Hg) and sulfur (S) attributable to the use of cinnabar or vermilion [29], also revealed by the quantitative analysis performed by PIXE and EBS that confirmed the stoichiometry of HgS. The presence of iron (Fe) suggests the possibility of a mixed compound with Fe-based materials such as earths or ochres (iron oxides) pigments. Differently for scroll no. 642, the red ink/pigment is not characterized by the presence of Hg but by quantities of Ca, even higher than the parchment itself. This may suggest the use of an organic compound mixed with iron oxides.

### 3.3.3. Yellow Ink/Pigment

The yellow ink areas of the unnumbered scroll and no. 643 are characterized by the presence of Fe (in lower quantities with respect to the red inks), probably linked to the use of earths or ochres (Fe oxides/hydroxides) (Figure 15). The presence of mercury (Hg) could be due, instead, to a contamination deriving from the migration of the red pigment during

the rolling/unrolling phases. This is generally true for all measuring points, in fact it is also revealed in the parchment. Similarly, Fe is also detected in the bare parchment (although with much lower yield) and its presence is possibly due to material migration as happens for Hg. It has to be noted, however, that it is common to detect Fe in parchments and its presence is due to the preparation process of the parchment itself [18,30].

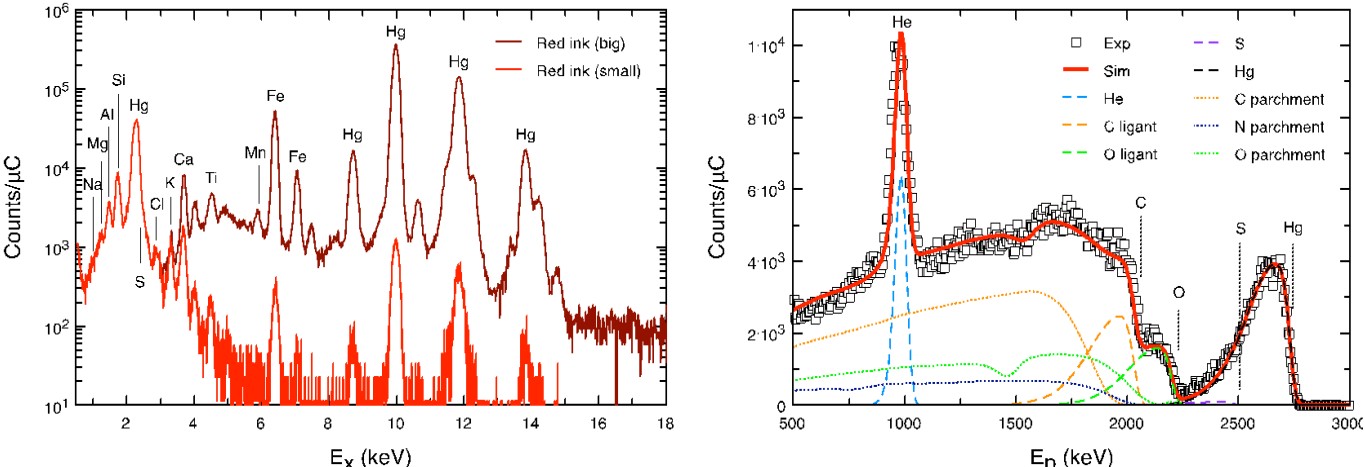

**Figure 14.** On the left, PIXE spectra of the red ink of the scroll without number obtained with the two SDDs, small and large area. On the right, EBS spectrum collected simultaneously, together with the SIMNRA simulation. Here, the contribution of the different elements to the simulation is also shown, and the distinction between carbon (C) and oxygen (O) in the parchment or in the ligand is shown. It has to be noted that He is not present in the sample itself, but is a "parasitic" element common to EBS spectra when measurements are performed in an external beam set-up under helium flow.

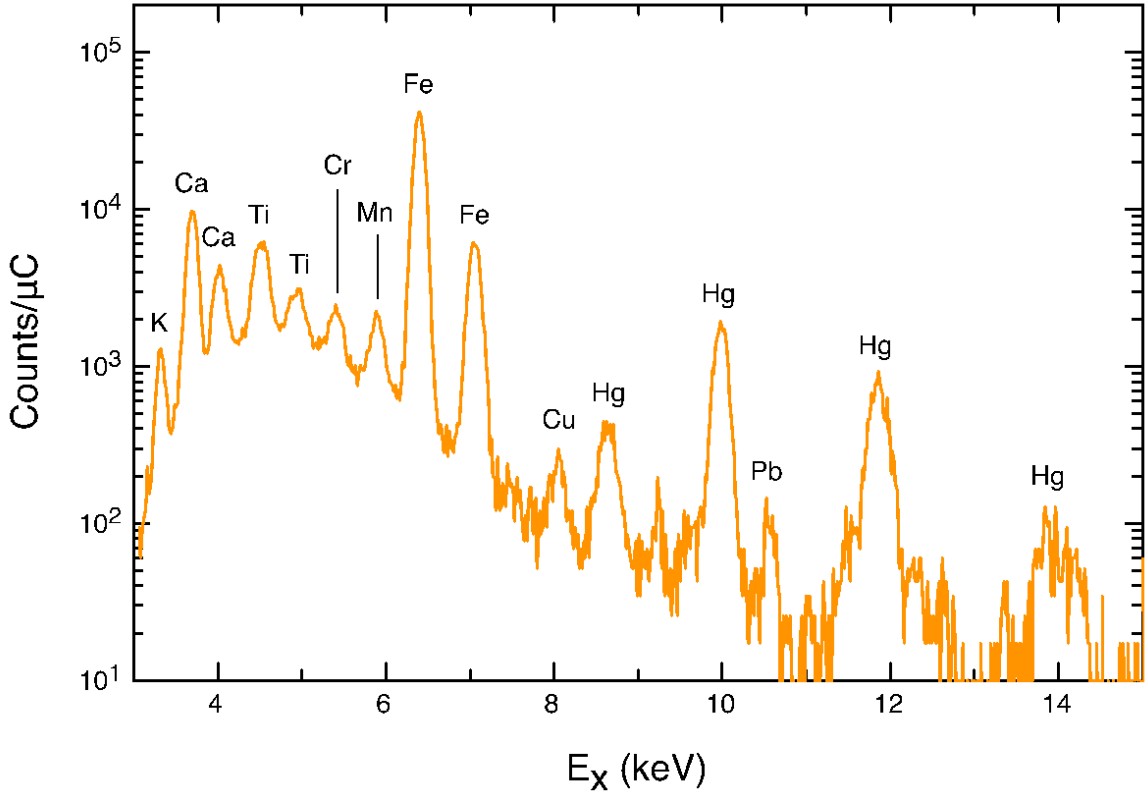

**Figure 15.** PIXE spectrum of the yellow ink on scroll 643 obtained with the large area SDD.

### 3.3.4. Brown Ink/Pigment

The brown ink of the scroll no. 642 shows the presence of different elements (Ca, Fe, K) from which, however, it is impossible to establish the pigment with certainty. It could be an organic compound combined with earths or ochres.

## 4. Discussion

The combined approach of imaging and spectroscopic analysis employed in this work has allowed us to characterize the composition of the original pigments and inks used and to obtain further interesting information with respect to what is present in the literature.

The black inks probably have a mixed composition consisting of the union of carbonaceous pigments, probably carbon black, and metal-gallic inks. The use of mixed-composition inks was quite common in that geographical area and was previously described for Egyptian papyrus documents from the Byzantine period [24]. The ratio of these two mixed components is different in the four scrolls. More specifically, scrolls no. 642, no. 644, and the scroll without number can be recognized as a mainly carbon-based ink since, in the mentioned cases, the IRR shows the strong persistence of the writing also in the NIR spectral range. Nevertheless, the presence of Fe highlighted by IBA suggests the possibility of using a mixed combination of the two kinds of inks.

On the other hand, scroll no. 643 seems to have a different composition of the mixture. Its brownish color, typical for metal-gallic inks, and the response of the writing at the IRR suggest the use of an opposite combination of the blend with a mainly metal-gall ink composition with small percentages of carbon black.

Interestingly enough, the results obtained from the diagnostic analyses regarding the black ink composition differ from what is stated in the literature that reports using carbon-based ink (carbon black) as traditional in Ethiopia [2]. Nevertheless, numerous studies carried out on various Ethiopian artifacts report as not unusual the employment of mixed inks. It is worth mentioning here that the use of different analytical techniques was of paramount importance in the characterization of carbon-based pigments and inks. In fact, FTIR studies on this topic are quite limited: Tomasini et al. [31] found that pigments based on graphite, lampblack, and charcoal give very poor IR spectra, while other pigments based on the carbonization of organic materials (i.e., ivory of vine black) showed the intense band of phosphate in the 1020–990 $cm^{-1}$ spectral region, which unfortunately overlap with gum IR absorption. Other pigments, such as bitumen and bistre, showed strong bands in the 1600–1570 and 870 $cm^{-1}$ regions, which correspond to aromatic and carbonate vibrations, respectively. Both bands are described in other studies [22,24] and may correspond to what was observed in our samples (Figure 9).

The red inks and pigments of scrolls no. 643, no. 644, and of the scroll without number are based on cinnabar/vermilion (IBA revealed the presence of Hg and S) while those of scroll no. 642 were obtained from an organic compound. Furthermore, thanks to both IBA and ATR-FTIR spectroscopy analyses, the presence of earths/ochres (as suggested by the presence of silicates in the FTIR spectra and of Fe in the IBA analysis) has been detected on the red pigments of all the scrolls. The infrared technique could not confirm the presence of this pigment since it does not absorb in the investigated spectral range (from 4000 to 400 $cm^{-1}$).

Although cinnabar/vermilion has already been identified in some Ethiopian manuscripts, traditional recipes do not mention it among the palette employed by the artisans. This is probably because the presence of cinnabar mines in Ethiopia can be excluded since the existence of this pigment would have made it an exporting country. Still, no mention of this is found in the documentary sources [32]. Furthermore, recipes for the production of vermilion (the synthesized version of cinnabar) have never been attested in Ethiopia. The most plausible theory remains that this kind of pigment was imported into Ethiopia after the eighteenth century; the artisan preparation of red pigments that only involved the use of organic ingredients gradually fell into disuse and was replaced by imported commercial red dyes. Therefore, the presence of cinnabar/vermilion

inside the red inks of three of the four Ethiopian scrolls is consistent with what has just been argued, while the use of organic compound on scroll no. 642 would be, instead, a rarity. The presence of earths/ochres in the red inks is compatible with documented Ethiopian customs [33].

The yellow ink of the scroll without number seems to be based on earth/ochre with the addition of a plant-based resin well recognizable from the bands on the ATR-FTIR spectra. The yellow ink of scroll no. 643 is probably an earth/ochre due to the presence of Fe highlighted by the IBA. Considering that very few traditional recipes concerning the production of yellow inks have come down to us, it is impossible to hypothesize the nature of the resin used on this scroll.

The brown ink of scroll no. 642 could be an organic compound combined with an earth/ochre. In this case, the lack of traditional recipes makes it challenging to compare with the pigments used in Ethiopia.

The analyses determined that the inks' binder is of the rubbery type, and is therefore congruent with the binding substances included in the recipes [2].

Finally, the analyses carried out on the parchment supports of the scrolls confirmed the production procedures of the parchments in Ethiopia described in the literature, in particular the immersion in salt water prior to processing (presence of Cl), the application of calcite powder in the skiving phase (presence of carbonates), and the use of kaolin to prepare the parchment for writing (presence of alumina-silicates).

## 5. Conclusions

The complementarity of two fields such as scientific and historical-artistic studies is essential in understanding this artifact as a cultural asset in all its aspects and to determine the right approach to its conservation.

The research carried out here has made it possible to characterize the Ethiopian magic scrolls as original parchment handicrafts about which much is not yet known.

The study of the typical features of these artistic objects linked to Ethiopian history, religion, and ritual practices (superstitions and traditional spiritual seances), reveals information about the civilization they belong to. The scientific analyses performed to complete the research made it possible to investigate the chemical-physical characteristics of the artifacts. This is fundamental for several aspects: (1) it helps in the reconstruction of the techniques for making parchment artifacts according to the customs, technologies, and availability of the civilization and the origin territory; (2) it allows us to identify the composition of the inks and pigments of the written/graphic text. Specifically, the acquisition of this knowledge will make it possible to identify the most suitable cleaning, restoration, and care methods for interventions on the four studied artifacts so that the best conservation state of the object and of the memory heritage it contains is guaranteed, beyond the language barriers.

The studies carried out add an important element in the research concerning Ethiopian artifacts; the large number of data collected can constitute fertile ground for future further insights aimed at learning about the traditional productions of Ethiopia.

The presented research gave an important result also from the point of view of the experimental approach to this case study. The use of imaging techniques alongside nuclear investigation techniques is a valid scientific approach for a detailed surface and in-depth study of the parchment artifacts. The procedure followed for the characterization of the Ethiopian magic scrolls has taken advantage of the propensity for multidisciplinary collaboration in the interest of the study of cultural heritage assets by combining the capabilities of nuclear physics with those of materials science in favor of the humanities and of the technologies for the conservation of artistic objects.

**Author Contributions:** Conceptualization, M.V. and C.C.; Data curation, M.V.; Formal analysis, M.V. and C.C.; Investigation, M.V., D.B., M.C., C.D.R., M.D.F., A.M. and C.L.Z.; Methodology, M.V. and C.D.R.; Resources, C.D.R. and C.L.Z.; Supervision, M.V.; Writing—original draft, M.V. and C.C.; Writing—review and editing, M.V., N.O. and C.C. All authors have read and agreed to the published version of the manuscript.

**Funding:** The participation of C.C. has been possible thanks to the co-funding of the European Union FSE REACT-EU-PON Ricerca e Innovazione 2014–2020, DM 1062/2021.

**Data Availability Statement:** Not applicable.

**Acknowledgments:** The authors would like to thank Rita Capitani, restorer and conservator of the archival and bibliographic superintendency of Emilia Romagna and Riccardo Pedrini, director of the *Archivio storico della provincia di Cristo Re dei Frati Minori dell'Emilia Romagna* of Bologna (Italy).

**Conflicts of Interest:** The authors declare no conflict of interest.

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
