# Peer review of "The Ethiopian Magic Scrolls: A Combined Approach for the Characterization of Inks and Pigments Composition"

_heritage, doi:10.3390/heritage6020075_

Round 1

Reviewer 1 Report

The article “The Ethiopian magic scrolls: a combined approach for the characterization of inks and pigments composition” is focused on the analysis of the inks present on the Scrolls of Ethiopia which were made by the Christians of Ethiopia as a protection against diseases and demonic possessions. Through the use of complementary techniques such as infrared reflectography, ATR-FTIR spectroscopy, Ion Beam Analysis (IBA) they wanted to analyze the inks for contributing to expand the knowledge on Ethiopian magic scrolls and of their production.

The article is linear, no discrepancies emerged on what was argued. But the conclusions, although not mandatory, in my opinion are necessary to summarize the work carried out, to underline the added contribution given to the literature and the future perspectives of analysis. For this reason, from my point of view, it is necessary to make this integration.
Here are additional changes that could be made to the article:

In indicating the centuries some notations are Italian as on page 2 in which it is reported "...I-VII a.C."; since the article is in English, the modification is necessary by replacing the time indications with BC.

The reference page was inserted on page 2 in the insertion of the citation [2], it is better to insert this further specification in the references below, as was done for the rest of the references.

On page 4 in the "Imaging techniques" section it would be advisable to anticipate which techniques were used. Furthermore, the name of the instrument used was not indicated in the IR reflectography section and I would elaborate on its technical specifications.

Figure 2 has not been mentioned in any part of the article, since the images support the argument, it would be appropriate to contextualize it.

For the identification of inks/pigments for yellow and brown no supporting citations have been inserted, their insertion is requested.

In indicating the number of artefacts, two different notations are used: n. and no. For homogeneity and coherence with English, we ask for their replacement with no.

- Finally, for the points analyzed with the other techniques it would be interesting to visualize in which points the analyzes were carried out, as was done for the IR reflectographic image.

Author Response

General comments:

The article “The Ethiopian magic scrolls: a combined approach for the characterization of inks and pigments composition” is focused on the analysis of the inks present on the Scrolls of Ethiopia which were made by the Christians of Ethiopia as a protection against diseases and demonic possessions. Through the use of complementary techniques such as infrared reflectography, ATR-FTIR spectroscopy, Ion Beam Analysis (IBA) they wanted to analyze the inks for contributing to expand the knowledge on Ethiopian magic scrolls and of their production.

The article is linear, no discrepancies emerged on what was argued. But the conclusions, although not mandatory, in my opinion are necessary to summarize the work carried out, to underline the added contribution given to the literature and the future perspectives of analysis. For this reason, from my point of view, it is necessary to make this integration.

According to the reviewer’s suggestion the conclusion section has been added to the manuscript.

Minor comments:

- In indicating the centuries some notations are Italian as on page 2 in which it is reported "...I-VII a.C."; since the article is in English, the modification is necessary by replacing the time indications with BC.

We apologize to reviewer, the text has been corrected according to the suggestion.

- The reference page was inserted on page 2 in the insertion of the citation [2], it is better to insert this further specification in the references below, as was done for the rest of the references.

Thanks for the note. The error of carrying the full citation in the references paragraph has been corrected as you suggested.

- On page 4 in the "Imaging techniques" section it would be advisable to anticipate which techniques were used. Furthermore, the name of the instrument used was not indicated in the IR reflectography section and I would elaborate on its technical specifications.

The name of the instrument and other technical specifications have been added on page 5, lines 148-151: The infrared images were acquired by means of a a Sony Cyber-shot DSC-F828 camera with 2/3" (3.9x) CCD sensor and a resolution of 8.0 megapixels. The camera is combined with a Carl Zeiss 7.1-51.0mm f /2.0-2.8. Different filters (Hoya R72 Infrared Filter, B+W 093 RG830 Filter, Schott RG1000 Filter)”.

- Figure 2 has not been mentioned in any part of the article, since the images support the argument, it would be appropriate to contextualize it.

The missing reference to figure 2 has been added to lines 190-1 of the text.

- For the identification of inks/pigments for yellow and brown no supporting citations have been inserted, their insertion is requested.

An additional reference was added to the paragraphs 3.2.2 describing yellow and brown pigments, which are both earth pigments. The reference is the following: [27] Genestar, C.; Pons, C. Earth pigments in painting: characterisation and differentiation by means of FTIR spectroscopy and SEM-EDS microanalysis   . Anal. Bioanal. Chem. 2005, 382, 269-274. https://doi.org/10.1007/s00216-005-3085-8.

- In indicating the number of artefacts, two different notations are used: n. and no. For homogeneity and coherence with English, we ask for their replacement with no.

All the notations have been replaced with no.

- Finally, for the points analyzed with the other techniques it would be interesting to visualize in which points the analyzes were carried out, as was done for the IR reflectographic image.

A new figure (now indicated as Figure 2), where the points investigated by ATR-FTIR spectroscopy and IBA analysis has been added.

Reviewer 2 Report

The manuscript approach studies scrolls by Ion Beam Analysis, FTIR, and IRR image. This is a typology of artifact, which requires studies to expand knowledge. The methodology applied in the study is adequate. For publication, I suggest the following revisions.

- In the method section, describe how the IRR images were processed.

- Input the dimensions of the parchments in figure 1.

- The outstanding result of the work would be the black pigments, which were seen by IRR image. Why did FTIR see carbon bands? I believe that it is not possible to characterize by technique. This point should be deepened.

- In the result of Ion Beam Analysis, Fe appears in all spectra. This point can be better justified in the part of the results of the Ion Beam Analysis, because from the IRR image it is clear that the ink used in the black part is carbon. Wouldn't that be the element of the parchment support?

Good work.

Author Response

Minor comments:

- In the method section, describe how the IRR images were processed.

The paragraph 2.2.1 has been inserted in the text on page 5, lines 148-151 and 156-159.

- Input the dimensions of the parchments in figure 1.

The dimensions of the scrolls have been added in the caption of figure 1.

- The outstanding result of the work would be the black pigments, which were seen by IRR image. Why did FTIR see carbon bands? I believe that it is not possible to characterize by technique. This point should be deepened.

We agree with the referee that the study of pigments and inks based on carbon by means of FTIR spectroscopy is not easy. Stimulated by the referee’s comment, we deepened our bibliographic research on the topic and added a the following comment in the discussion section:It is worth mentioning here that the use of different analytical techniques was of paramount importance in the characterization of carbon-based pigments and inks. In fact, FTIR studies on this topic are quite limited: Tomasini et al. [31] Tomasini, E.; Siracusano, G.; Maier, M.S.  Spectroscopic, morphological and chemical characterization of historic pigments based on carbon. Paths for the identification of an artistic pigment. Microchem. J. 2012, 102, 28-37. https://doi.org/ 10.1016/j.microc.2011.11.005] found that pigments based on graphite, lampblack and charcoal give very poor IR spectra, while other pigments based on the carbonization of organic materials (i.e. ivory of vine black) showed the intense band of phosphate in the 1020-990 cm-1 spectral region, which unfortunately overlap with gum IR absorption. Other pigments, such as bitumen and bistre, showed strong bands in the 1600-1570 and 870 cm-1 regions, which correspond to aromatic and carbonate vibrations, respectively. Both bands are described in other studies [22, 24] and may correspond to what observed in our samples (Figure 9)”.

The text is contained in lines 484-493

- In the result of Ion Beam Analysis, Fe appears in all spectra. This point can be better justified in the part of the results of the Ion Beam Analysis, because from the IRR image it is clear that the ink used in the black part is carbon. Wouldn't that be the element of the parchment support?

We thank the referee for the remark, Fe i is most likely due to the preparation procedure of the parchment. We have added a comment on this with two related references [18,30] on page 16, lines 45-46.

Round 2

Reviewer 1 Report

For the paper “The Ethiopian magic scrolls: a combined approach for the characterization of inks and pigments composition” have been taken into consideration all the suggested changes, especially the integration of images and the addition of conclusions has made the article more complete and comprehensive.
The authors have responded and integrated to all requests made. I recommend the publication of the paper in its current form.

Reviewer 2 Report

All suggested comments were implemented by the authors. I am in favor of accepting the article in its current format.